# Newborn Screening for X-Linked Adrenoleukodystrophy: Past, Present, and Future

**DOI:** 10.3390/ijns8010016

**Published:** 2022-02-18

**Authors:** Ann B. Moser, Elisa Seeger, Gerald V. Raymond

**Affiliations:** 1Kennedy Krieger Institute, Baltimore, MD 21205, USA; mosera@kennedykrieger.org; 2ALD Alliance, Brooklyn, NY 11218, USA; elisa@aldalliance.org; 3Department of Genetic Medicine, Johns Hopkins Hospital, Baltimore, MD 21287, USA

**Keywords:** adrenoleukodystrophy, newborn screening, dried blood spots, adrenal insufficiency, cerebral adrenoleukodystrophy

## Abstract

Newborn screening for X-linked adrenoleukodystrophy began in New York in 2013. Prior to this start, there was already significant information on the diagnosis and monitoring of asymptomatic individuals. Methods needed to be developed and validated for the use of dried blood spots. Following its institution in New York, its acceptance as a disorder on the Recommended Uniform Screening occurred. With it has come published recommendations on the surveillance and care of boys detected by newborn screening. There still remain challenges, but it is hoped that with periodic review, they may be overcome.

## 1. Introduction

Newborn screening for X-linked adrenoleukodystrophy (ALD) was added to the Recommended Uniform Screening Panel in February 2016. The implementation of this screening continues to expand in the United States and is being considered in other countries [1,2,3,4,5,6,7,8,9]. It is an opportune time to evaluate the development, performance, and the future of the endeavor.

ALD is a progressive condition that is due to defects in the peroxisomal ATPase binding cassette protein encoded by the gene *ABCD1* [10]. Affected individuals have an elevation in saturated very long chain fatty acids (VLCFA) in all body tissues, but this accumulation primarily affects the nervous system and adrenal cortex. This increase in saturated VLCFA, especially hexacosanoic acid (C26:0), is diagnostic for many peroxisomal disorders and is implicated in the pathogenesis. The clinical and pathologic characteristics have been discussed in recent reviews. The diagnosis and treatment of adrenal insufficiency (Addison disease) and childhood cerebral disease have been established [10,11,12,13,14,15,16,17].

## 2. Rationale and Development of Newborn Screening

With the development of a biochemical assay, extended family screening could be performed, expanding the population of asymptomatic males. Monitoring for adrenal insufficiency and MRI imaging surveillance for cerebral disease could be performed in the at-risk population. Dubey et al. demonstrated that ACTH levels were abnormal in males at very early ages prior to overt adrenal disease [11]. Similarly, clinical neurologic disease was preceded by neuroimaging changes. This also led to the demonstration of the benefit of early intervention using HSCT, prior to advanced disease on MRI [12]. Unfortunately, boys still came to attention with either catastrophic injury from an adrenal crisis or advanced cerebral disease precluding therapy.

It was against this background that the first work to establish universal screening for ALD began. Using tandem mass spectroscopy, a methodology to examine the C26-lysophosphatidyl choline fraction from dried blood spots was developed [13,14]. It was shown that it could be adapted for automated measurement and performed in conjunction with other common newborn assays [2,15,16]. Clinical studies demonstrated that it was reliable and effective [17].

## 3. Initiation and Implementation

In 2011, Aidan Seeger, a 6-year boy from Brooklyn, developed vision and concentration issues. Eventually diagnosed with childhood cerebral ALD, he died 11 months later.

His parents, upon learning that a method for newborn screening for ALD was available but was not in use, made it their mission to have it implemented in New York. Mounting a broad campaign of public support and direct work with the New York legislature, they were able to see Aidan’s Law passed and signed on 31 March 2013, with implementation of newborn screening in December 2013. The New York newborn screening program undertook a rapid, effective program to establish testing and follow-up in the state using the established metabolic centers [3].

With the establishment in New York, stakeholders petitioned for its inclusion on the Recommended Uniform Screening Panel (RUSP), which occurred in February 2016.

## 4. Discussion

Newborn screening for ALD is now established in 24 states and the District of Columbia, although there remain issues—some expected and some unexpected [3,8,9]. The main issue remains the disparity of state-to-state screening. A child born in Pennsylvania or Washington, D.C. will have the opportunities provided by screening for ALD, but a child born in Maryland will not. This geographical inequity continues to plague many disorders detected by newborn screening even when they are listed on the RUSP.

Issues of methodologies and harmonization of cut-offs persist. The placement of screening cut-offs has clear implications for the number of follow-up samples required. The competing interests of minimizing referrals for secondary testing without accepting false negatives is not unique to ALD. Some state programs provide additional biochemical and genetic testing to improve their ability to refer only those individuals who have a high likelihood of disease. Multi-tiered biochemical screening has demonstrated itself to be a cost-efficient compromise but necessitates additional equipment and expertise.

Genetic testing, either in the state programs or by the genetic centers, has brought with it other concerns. There are many disease-causing variants of *ABCD1* [18]. An international curated database maintained by Dr. Stephen Kemp in the Netherlands (www.adrenoleukodystrophy.info, accessed on 1 January 2022) currently lists 3401 variants and about a third are non-recurrent. In the past, there were a small number of genetic labs providing this DNA testing, and they would have access to family pedigrees. Segregation information tied to biochemical data was readily accessible to assist in the determination of whether a variant was disease-causing or not. Now with many more state and commercial genetic labs providing this testing, the labeling of variants as of uncertain significance has increased exponentially, creating ambiguity at a point where it is not desirable: an infant who may or may not have a disorder with significant implications for surveillance and potential disease.

Detection of carrier females continues to be a debated point. The balance between universal screening and intrusion into the autonomy of an asymptomatic female heterozygote who does not develop any childhood disease is still not settled. Newborn detection may identify and potentially a male relative who was not afforded the advantage of newborn screening. Strategies for testing only specimens from males are being explored in a pilot program [1].

Other peroxisomal disorders may be identified, including in children with the Zellweger spectrum disorder and single-enzyme defects in peroxisomal beta oxidation. It was not expected that a subset of children with Aicardi–Goutieres syndrome [19,20] would be flagged. This has brought an added dimension in the evaluation of the child who is being seen in follow up.

The goal of early identification remains to keep the identified boy healthy and productive, so it is very important that the infant and his family should receive diagnoses and counseling in a timely fashion. Guidelines for monitoring the boy diagnosed with ALD by newborn screening have been published and include expedited referral to an endocrinologist and a schedule for MRI monitoring [21,22]. It is essential that the initial encounter with the family be in a program that has experience with ALD, and it is desirable that follow-up be provided a multidisciplinary team that is able to coordinate all aspects of care. For those metabolic genetic programs that are unfamiliar with ALD, there are programs and providers willing to pair with them for education and consultation.

The collection of long-term outcome data is essential for demonstrating and improving the quality of care of individuals affected by this condition, but most state newborn follow-up programs are limited in scope and woefully underfunded. In addition, there are significant barriers to data sharing among these public health institutions. There is clearly a role here for the federal government and an institution such as the CDC to improve data collection through harmonization.

## 5. Conclusions

Newborn screening for ALD has been successfully instituted in 24 states and is identifying individuals accurately and efficiently. It is appropriate in this special edition to review the experience and highlight methods that states are using to diagnose and follow up on individuals detected.

As we accumulate information it will be, of course, imperative that we continue to review and acknowledge what is working, but also, just as importantly, establish what is not and correct it. This is clearly owed to those who have been affected by this disease and may or may not have benefited from early detection.

## Data Availability

Not applicable.

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
