# Peer review of "Newborn Screening for X-Linked Adrenoleukodystrophy: Past, Present, and Future"

_2409-515X, 2022, doi:10.3390/ijns8010016_

Round 1

Reviewer 1 Report

Moser and colleagues attempt to tell the story of X-ALD’s addition to newborn screening, review the current status of newborn screening for the disorder, and challenges that remain.  This is an important story.  However, the manuscript suffers from lack of linearity and grammatical errors, rendering it unpublishable in its current version.  In addition, the most pertinent information does not appear until page 3 of this short editorial.

Major:

Grammar needs to be improved throughout, much of the text is conversational: 

Abstract, sentence 1.  “…beginning began almost a decade before.”

Abstract, sentence 2 is an incomplete sentence.

The last sentence of the abstract is awkwardly written.

Introduction, sentence 1 is a run on sentence.

Introduction, sentence 2: It is an opportune time, this reviewer agrees, it does not appear to be.

Introduction, lines 33-34, semicolons are not needed.

Introduction, paragraph 4, line 40: “inattention” is listed twice.  This sentence requires a citation, for example: Raymond et al 2019

Introduction, paragraph 4, last sentence: “even” is used twice

And so forth..

Additional comment: The perspective of the storyteller switches between third person, fourth person indirect point of view, and the first person.  The manuscript should be written in the third person, directly pointing out challenges and how the events transpired.

Other comments: The authors spend 1.5 pages reviewing the disorder.  This reviewer suggests focusing on the historical development of the NBS assay (nicely done in section 2), and the state of NBS and follow up as it stands today.   More importantly, the most pertinent and up-to-date section of this paper is under “discussion”, and should be expanded.  It is this reviewer’s suggestion that the current and forward-facing challenges with X-ALD screening should be the bulk of the manuscript.    

Author Response

We acknowledge the reviewer's comments. The introduction has now been substantially shortened so that many of the comments are now moot.

Major:

Grammar needs to be improved throughout, much of the text is conversational: 

Abstract, sentence 1.  “…beginning began almost a decade before.” Corrected

Abstract, sentence 2 is an incomplete sentence. Corrected

The last sentence of the abstract is awkwardly written. Changed

Introduction, sentence 1 is a run on sentence. Corrected

Introduction, sentence 2: It is an opportune time, this reviewer agrees, it does not appear to be. Not certain what the reviewer is stating, but it is edited

Introduction, lines 33-34, semicolons are not needed. This section has been removed

Introduction, paragraph 4, line 40: “inattention” is listed twice.  This sentence requires a citation, for example: Raymond et al 2019 This again has been edited out in the revision

Introduction, paragraph 4, last sentence: “even” is used twice Revised

Additional comment: The perspective of the storyteller switches between third person, fourth person indirect point of view, and the first person.  The manuscript should be written in the third person, directly pointing out challenges and how the events transpired.

Revised although the authors are confused by the statement fourth person. I review of grammar indicates that there are only three persons in the English and most Indo-European languages. It is noted that a fourth person exists in some Native American languages and others, but again it has been revised

Other comments: The authors spend 1.5 pages reviewing the disorder.  This reviewer suggests focusing on the historical development of the NBS assay (nicely done in section 2), and the state of NBS and follow up as it stands today.   More importantly, the most pertinent and up-to-date section of this paper is under “discussion”, and should be expanded.  It is this reviewer’s suggestion that the current and forward-facing challenges with X-ALD screening should be the bulk of the manuscript.   

The introduction has been shortened and revised.

Since this was to be a short introduction to a special edition on newborn screening for adrenoleukodystrophy, it was initially our desire to reflect on some of the history of the development and since all three authors were intimately involved to provide some of the personal story. The reviewer prefers a dispassionate rendition, so with one exception - the exceptional story of the efforts of the Seeger family - we have complied

Reviewer 2 Report

This commentary is useful and, as the authors suggest, written at an opportune time to reflect upon progress with the implementation of newborn screening for ALD in the US.

There are some minor issues with English usage and some errors that need correction:

  • On p1, line 11 ‘While there was already significant information on diagnosis and monitoring of asymptomatic individuals. Methods needed to be developed …………….’ Should be corrected to: ‘While there was already significant information on diagnosis and monitoring of asymptomatic individuals methods needed to be developed …………….’ 
  • On p1, line 13 should be corrected to ‘the Recommended Uniform Screening Panel’
  • On p1, line 22, should perhaps be ‘six years ago’ as it was Feb 2016 that ALD was included within the RUSP
  • On p1, line 40, ‘inattention’ is repeated.
  • On p2, line 62, there is an extra space before the word ‘generally’
  • On p3, line 138, the text is confusing around ‘,an infant who may or may not have a disorder’ and this should be re-phrased.
  • On p3, line 144, should read ‘receive diagnosis’
  • On p4, line 149, it should read – ‘multidisciplinary team’

With these minor revisions and corrections, I would recommend publication.

Author Response

We thank the reviewer for their comments. In the revision, all of their points have been addressed

Round 2

Reviewer 1 Report

None